# Anti-TNF Alpha and Risk of Lymphoma in Rheumatoid Arthritis: A Systematic Review and Meta-Analysis

**DOI:** 10.3390/medicina60071156

**Published:** 2024-07-17

**Authors:** Ahmad A. Imam

**Affiliations:** Internal Medicine Department, College of Medicine, Umm Al-Qura University, Makkah 24382, Saudi Arabia; aaimam@uqu.edu.sa

**Keywords:** anti-TNF, systematic review, lymphoma, conventional treatment

## Abstract

*Background and Objectives:* Anti-tumor necrosis factor-alpha (TNF-α) agents are effective in treating rheumatoid arthritis (RA) but may entail a risk of lymphoma due to TNF-α’s role in immune surveillance. This systematic review and meta-analysis assesses the risk of lymphoma in patients with RA treated with anti-TNF agents versus patients treated with methotrexate and/or a placebo. *Materials and Methods:* The Cochrane Database of Systematic Reviews, Cochrane Central Register of Controlled Trials, Embase, PubMed, and Google Scholar were systematically searched for relevant literature. Data were extracted and analyzed to determine risk ratios (RRs) and 95% confidence intervals (CIs), with heterogeneity assessed using I^2^ statistics. Methodological quality and risk of bias were assessed using the Cochrane Risk of Bias tool for randomized controlled trials (RCTs) and the Newcastle–Ottawa Scale for observational studies. *Results:* The search yielded 932 articles, 13 of which were retained for qualitative review and 12 for quantitative synthesis. Overall, the studies reviewed included 181,735 participants: 3772 from six RCTs and 177,963 from seven observational studies. The meta-analysis of RCTs revealed no significant difference in the risk of lymphoma between patients receiving anti-TNF-α therapy and patients on conventional treatments, with an overall RR of 1.43 (95% CI: 0.32–5.16) and I^2^ of 0%. Conversely, observational studies showed some variability, with an overall RR of 1.43 (95% CI: 0.59–3.47) and significant heterogeneity (I^2^ = 95%), whereas others indicated a potentially elevated risk of lymphoma in specific subgroups but had inconsistent results. *Conclusions:* The systematic and meta-analysis revealed no significant difference in the risk of lymphoma for patients with RA treated with anti-TNF-α agents versus conventional therapies. However, given the limitations of the studies included, additional research is needed to validate the results and explore potential risk factors contributing to the development of lymphoma in patients with RA.

## 1. Introduction

Rheumatoid arthritis (RA) is a chronic autoimmune disease characterized by inflammation of the joints that leads to pain, swelling, stiffness, and progressive joint damage, and it consequently imposes significant limitations on patients’ daily activities [1]. To combat that progressive deterioration, a multitude of anti-rheumatic medications have been developed and employed to modify the course of the disease and alleviate symptoms. Among those therapeutic options, anti-tumor necrosis factor-alpha (TNF-α) agents have emerged as pivotal components in the management of RA [2,3,4]. TNF-α, a key proinflammatory cytokine implicated in the pathogenesis of RA, serves as the primary target of those biologic therapies. By inhibiting the action of TNF-α, drugs such as infliximab, adalimumab, certolizumab pegol, golimumab, and etanercept suppress inflammation and impede the disease’s progression in patients with RA.

However, because the blockade of TNF-α also plays a crucial role in modulating immune responses against B cell lymphomas, the long-term use of the agents raises concerns regarding the associated potential risk of developing lymphoma, a diverse group of malignancies arising from lymphocytes [5,6]. Briefly put, lymphoma is a group of cancers originating in the lymphatic system, a part of the body’s immune system that includes lymph nodes, the spleen, the thymus, and bone marrow, among other components. Lymphomas can be broadly categorized into two types: Hodgkin lymphoma (HL) and non-Hodgkin lymphoma (NHL) [1,2,3,4]. In particular, concerns regarding anti-TNF-α therapy’s safety profile, including its potential association with an increased risk of lymphoma, partly stem from the fact that patients with RA already exhibit a moderately higher risk of lymphoma than the general population due to factors such as chronic inflammation, immune dysregulation, and genetic predispositions. The inhibition of TNF-α by anti-TNF-α therapy may disrupt immune surveillance mechanisms, which may theoretically foster lymphomagenesis among patients with RA [7,8].

The literature on the association between anti-TNF-α therapy and the risk of lymphoma among patients with RA is characterized by a mix of observational studies, randomized controlled trials (RCTs), meta-analyses, and systematic reviews, which together show conflicting findings that sustain ongoing debate. Whereas some studies have suggested a modestly increased risk of lymphoma associated with TNF-α inhibitors, others have reported no significant association. Moreover, heterogeneity in study designs, patient populations, treatment regimens, and follow-up durations complicates efforts to draw definitive conclusions [9,10,11].

Given the described risk’s clinical significance and the associated implications for treatment decision making, there is a pressing need to systematically synthesize existing evidence about the relationship between anti-TNF-α and the risk of lymphoma. In response, this systematic review addresses that critical gap in the knowledge by examining the available evidence regarding the potential link between anti-TNF-α medications and the risk of all types of lymphoma among patients with RA. By synthesizing data from diverse studies and employing robust statistical analyses, the review seeks to clarify whether using anti-TNF-α agents confers an increased risk of lymphoma compared with conventional treatment modalities.

## 2. Materials and Methods

### 2.1. Search Strategy and Protocol

To ensure a comprehensive, systematic approach to identifying articles to review, a thorough search of multiple databases was conducted. The Cochrane Database of Systematic Reviews, Cochrane Central Register of Controlled Trials, Embase, PubMed, and Google Scholar [12,13,14,15] were systematically searched using a predefined search strategy to identify relevant articles. The strategy employed a combination of keywords and MeSH terms related to rheumatoid arthritis, anti-TNF-alpha, and lymphoma, namely “rheumatoid arthritis OR anti-TNF-α OR infliximab OR adalimumab OR etanercept AND lymphoma OR hematological malignancy.” Moreover, to minimize publication bias, the reference lists of retrieved articles were manually screened and reviewed for additional relevant literature. The Preferred Reporting Items for Systematic Reviews and Meta-Analyses (PRISMA) checklist was utilized to identify and select the literature included in the systematic review and meta-analysis.

### 2.2. Selection Criteria

Primary screening was conducted using the titles and abstracts of identified articles to assess their eligibility based on predetermined inclusion criteria. Articles were included if they (1) presented original research in peer-reviewed journals, (2) investigated the association between anti-TNF-α therapy and the risk of lymphoma in patients with RA, (3) were observational studies (i.e., cohort, case–control, or nested case–control) or RCTs, and (4) were written in English. Conversely, articles were excluded if they did not meet those criteria or if they (1) involved patients less than 18 years old, (2) examined anti-TNF-α’s risk of lymphoma in relation to other rheumatological diseases, or (3) did not include lymphoma as an outcome measure.

### 2.3. Data Extraction

Data were extracted from the included articles using a standardized data extraction form. Extracted data included characteristics of the article or study (i.e., author, year of publication year, and study design), participants’ characteristics (i.e., sample size and demographics), details about the intervention (e.g., type and duration of anti-TNF-α therapy), outcomes (i.e., incidence of lymphoma), and measures of association (i.e., risk ratios [RRs], hazard ratios [HRs], and odds ratios [ORs]) with corresponding 95% confidence intervals (CIs). The process was performed systematically to ensure accurate, comprehensive data extraction from each article included.

### 2.4. Quality Assessment

The methodological quality and risk of bias of the included studies were assessed using appropriate tools tailored to the study design: the Newcastle–Ottawa Scale for observational studies and the Cochrane Risk of Bias tool for RCTs. Key domains evaluated included study design, participant selection, comparability of groups, exposure and outcome ascertainment, and statistical analysis. Studies were categorized as having high, moderate, or low quality based on the overall risk of bias [16].

### 2.5. Data Synthesis and Meta-Analysis

Quantitative data synthesis was performed using appropriate statistical methods, including meta-analysis when feasible. Pooled effect estimates (e.g., risk ratios) and corresponding 95% CIs were calculated using random-effects models depending on the heterogeneity between studies. Heterogeneity was assessed using the I^2^ statistic, with values exceeding 50% indicating substantial heterogeneity. Publication bias was evaluated using funnel plots and Egger’s test if a sufficient number of studies were included. ReviewManager software (version 5.4), supported by Cochrane Reviews, was utilized for conducting the meta-analysis, which involved calculating CIs and *p* values and assessing heterogeneity between studies.

### 2.6. Ethical Considerations

Because the research presented herein involved the synthesis and analysis of data in published literature, ethics approval was not required.

## 3. Results

The literature search initially yielded 932 articles that met the inclusion criteria. After the exclusion criteria were applied, 919 studies were excluded, meaning that 13 articles were retained for qualitative review and 12 for quantitative synthesis (Figure 1). The final selection included six RCTs [17,18,19,20,21,22] and seven observational studies [23,24,25,26,27,28,29], all conducted between 2004 and 2018 in places such as the United Kingdom, the United States, Europe, North America, Australia, and Sweden, for a total of 181,735 participants. Baseline characteristics and the incidence of lymphoma in the RCTs and observational studies are detailed in Table 1 and Table 2, respectively.

### 3.1. Study Characteristics

#### 3.1.1. RCTs: Qualitative Findings

In a qualitative synthesis of six RCTs, conducted in places such as the United Kingdom, the United States, Europe, North America, Australia, and Sweden and involving 3772 participants overall, the efficacy and safety of anti-TNF-α agents in patients with RA were assessed (Table 1). In one RCT, Breedveld et al. (2006) demonstrated that combination therapy with adalimumab and methotrexate (MTX) significantly improved RA symptoms and clinical remission, with adverse event profiles comparable to MTX alone, but observed no cases of lymphoma in the anti-TNF-α group and only one case in the conventional therapy group [17]. Westhovens et al. (2006), by comparison, found that infliximab plus MTX was tolerated well but that the higher dose increased risks of serious infection, with one case of lymphoma in the anti-TNF-α group and none in the MTX group [18]. Two years before those RCTs, Keystone et al. (2004) showed that adalimumab significantly inhibited joint damage and was tolerated well, with one case of lymphoma in the anti-TNF-α group and none in the MTX group [19]. Beyond that, Kay et al. (2008) reported that golimumab plus MTX achieved significant clinical responses without cases of lymphoma in either group [20]. Emery et al. (2013) observed significant clinical and radiographic benefits with golimumab plus MTX over a two-year period, with one case of lymphoma in the high-dose arm and none in the MTX group [21]. Last, Smolen JS et al. (2012) confirmed sustained improvements in RA with injections of golimumab and consistent safety, with multiple cases of lymphoma in the high-dose group over an extended follow-up [22]. Overall, those studies highlight the efficacy of anti-TNF-α therapies in improving outcomes in RA, albeit with some incidence of lymphoma observed that requires careful long-term monitoring.

#### 3.1.2. Observational Studies: Qualitative Findings

In a comprehensive analysis of seven observational studies on treatments for RA, with a total population of 177,963 patients with RA, the risk of lymphoma among patients receiving anti-TNF-α agents was compared with the risk of ones receiving conventional therapy (Table 2). Most recently, Calip et al. (2018), who studied 947 patients with RA in the United States, found a nearly twofold increased risk of NHL among anti-TNF-α users (OR = 1.93; 95% CI: 1.16–3.20), particularly with TNF fusion proteins such as etanercept (OR = 2.73; 95% CI: 1.40–5.33) [23]. The year prior, in the United Kingdom, Mercer et al. (2017) reported no significant difference in the risk of lymphoma between TNFI-treated and biologic-naïve patients (HR = 1.00; 95% CI: 0.56–1.80) [24]. In Sweden, several years earlier, Askling et al. (2008) observed a relative risk of 1.35 (95% CI: 0.82–2.11) for lymphoma in patients with RA treated with anti-TNF-α versus ones naïve to anti-TNF-α, with a higher risk than the general population (RR = 2.72; 95% CI: 1.82–4.08) [25]. Meanwhile, in the United States, Wolfe et al. (2007) found no overall increased risk of cancer with biologics (OR = 1.00; 95% CI: 0.8–1.2), despite an increased risk of non-melanotic skin cancer and melanoma [26]. In Sweden two years earlier, Geborek et al. (2005) observed lymphoma in their study’s anti-TNF-α group (RR = 11.5; 95% CI: 3.7–26.9) [27]. Also in Sweden that year, Askling et al. (2005) reported a threefold increased risk of lymphoma (standardized incidence ratio [SIR] = 2.9) in patients treated with a TNF-α antagonist compared with the general population [28]. Last, in 2004 in the United States, Wolfe et al. found an SIR for lymphoma of 2.9 (95% CI: 1.7–4.9) among biologic users [29]. Taken together, those findings highlight the complex relationship between anti-TNF-α therapy and the risk of lymphoma and underscore the need for careful patient monitoring and risk assessment in treating RA.

### 3.2. Results of Quantitative Synthesis 

Five studies were included in the quantitative synthesis of RCTs because Kay et al. (2008) could not be included due to having zero events in both groups, which resulted in non-estimable results. Therefore, the findings from the study were limited to qualitative synthesis [20]. The RRs and their corresponding 95% CIs are presented in Figure 2.

The overall RR across all RCTs was 1.28 (95% CI: 0.32–5.16). Heterogeneity analysis showed no significant variability among the studies, with a tau-square score of 0.00, a chi-square score of 3.01, 4 degrees of freedom (*dF*; *p* = 0.56), and an I^2^ score of 0%, all of which suggest no heterogeneity among studies. The test for the overall effect resulted in a *Z* score of 0.34 (*p* = 0.73), which indicates no statistically significant difference. Therefore, the findings indicate no significant difference in the risk of events between patients receiving anti-TNF-α therapy and those receiving conventional therapy.

By contrast, a meta-analysis of seven observational studies, which involved computing the RRs and their corresponding 95% CIs, is presented in Figure 3. The overall RR across those studies was 1.43 (95% CI: 0.59–3.47), which suggests a 43% greater risk of events in the anti-TNF-α therapy group than in the conventional therapy group. However, the overall CI included 1, indicating no statistically significant difference. There were 206 events in the anti-TNF-α group and 826 events in the conventional therapy group. Heterogeneity analysis revealed significant variability among the observational studies, with a tau-square score of 1.29, a chi-square score of 118.51, 6 *dF* (*p* < 0.00001), and an I^2^ score of 95%, which together indicate substantial heterogeneity. The test for an overall effect resulted in a Z score of 0.80 *(p* = 0.42), which is not statistically significant. Therefore, despite variability among the studies, the findings suggest no statistically significant difference in the risk of events between patients receiving anti-TNF-α therapy and those receiving conventional therapy.

The meta-analysis also involved examining funnel plots for both the RCTs and observational studies in order to assess publication bias. The funnel plot for the RCTs was symmetrical, with results widely distributed within the plot, which suggests a lack of publication bias. Conversely, the funnel plot for observational studies was asymmetrical, with four results lying outside the plot, which indicates the potential presence of publication bias. However, given the small number of studies reviewed, making a conclusive determination regarding publication bias remains problematic. Figure 4 and Figure 5 illustrate those findings for the RCTs and observational studies, respectively.

### 3.3. Risk of Bias Assessment

In Figure 6A, traffic light plots illustrate the domain-level judgments for each individual result, whereas Figure 6B presents weighted bar plots depicting the distribution of risk-of-bias judgments within each bias domain. Four studies—Breedveld et al. (2006), Westhovens et al. (2006), Kay et al. (2008), and Smolen et al. (2012)—were assessed as having a low risk of bias across all domains, for an overall low risk of bias [17,18,20,22]. Keystone et al. (2004) showed some concerns in the domain related to missing outcome data (i.e., D3), for an overall assessment of “some concerns” [19]. Emery et al. (2013) presented a low risk of bias in all domains but some concerns in deviations from intended interventions (i.e., D2) and selection of the reported result (i.e., D5), culminating in an overall assessment of “some concerns” [21]. The Newcastle–Ottawa Scale was used for observational studies, and all studies included were high in quality for the domains of selection, comparability, and outcomes.

## 4. Discussion

The review included both RCTs and observational studies in order to enhance the precision of information concerning the risk of lymphoma in patients with RA undergoing treatment with anti-TNF-α agents compared with patients treated with methotrexate and/or a placebo. The total population encompassed in the review amounted to 181,735 individuals, with 3772 participants in RCTs and 177,963 in observational studies [17,18,19,20,21,22,23,24,25,26,27,28,29]. Such a substantial sample included a diverse range of patients’ characteristics, which strengthens the generalizability and external validity of the findings. Moreover, the large and varied data set facilitated more accurate statistical inferences than could be achieved by analyzing a single RCT or observational study only.

Although other systematic reviews have measured the risk of malignancies and serious infections among patients treated with anti-TNF-α, most have focused exclusively on RCTs [17,18,19,20,21,22]. Another meta-analysis examined the effect of only one type of biological treatment (e.g., etanercept) or included articles published by pharmaceutical companies that manufactured the biological medications [30].

The RCTs included in our analysis demonstrated no statistically significant difference in the risk of lymphoma between patients receiving anti-TNF-α therapy and those on conventional treatments (e.g., methotrexate). The lack of significant association was consistent across different demographic and treatment groups, as evidenced by the RRs and their associated 95% CIs. The homogeneity of the RCTs, indicated by an I^2^ value of 0%, supports the reliability of those findings [17,18,19,20,21,22].

Both Breedveld et al. [17] and Westhovens et al. [18] have provided robust evidence based on well-structured trials with sufficient follow-up periods. Their studies adhered to rigorous protocols that included washout periods before commencing treatment in order to account for latency-related issues in diagnosing lymphoma, and those protocols enhanced the validity of their results [17,18].

By contrast, the observational studies presented a rather complex picture. Although most such studies indicated no significant association between anti-TNF-α therapy and risk of lymphoma, with most CIs crossing the “no effect” line, two studies revealed a significant association [23,24,25,26,27,28,29]. For instance, Calip et al. found a nearly twofold increased risk of NHL in anti-TNF-α users, particularly with TNF fusion proteins such as etanercept [23]. Similarly, Geborek et al. reported a substantial RR of 11.5, although the risk reduced significantly when accounting for latency periods [27].

The substantial heterogeneity observed among the observational studies (I^2^ = 95%) underscores the variability in study designs, populations, and methodologies, which together complicate direct comparisons and definitive conclusions [31,32]. Such heterogeneity could be attributed to differences in patient populations, follow-up durations, and definitions of exposure and outcome, among other factors.

### 4.1. Strengths and Limitations

The RCTs included in the review exhibit several strengths, including a well-defined population sample, clear randomization, blinding processes, and a multicenter approach. For instance, Breedveld et al.’s PREMIER study was conducted across 133 centers in Australia, Europe, and North America [17], and Keystone et al.’s study involved 89 centers in the United States and Canada [19]. The statistical power and sample size calculations were clearly defined in Keystone et al.’s [19] and Westhovens et al.’s [18] studies, which enhanced the reliability of their findings. However, Breedveld et al. did not clearly outline the calculation of their sample size, which may limit the accuracy of their results [17].

The duration of the studies varied as well, with Smolen JS et al.’s lasting 40 months, and Kay et al.’s lasting 12 months. The longer follow-up duration in Smolen JS et al.’s study is critical for assessing rare outcomes such as lymphoma, which require significant time to develop [17,18,19,20,21,22].

Breedveld et al. focused on patients with RA with a disease duration of less than 3 years, thereby reducing the confounding effect of disease duration and severity on the risk of lymphoma [17]. By contrast, Keystone et al. and Westhovens et al. included patients with a median disease duration of 10 years or those resistant to disease-modifying antirheumatic drugs (i.e., DMARDs), which complicated determining whether lymphoma was caused by the disease or anti-TNF-α therapy [18,19].

Some details in those studies are worth noting. For example, in Keystone et al.’s study, only one patient developed lymphoma in the adalimumab group, but the authors did not specify which treatment regimen (i.e., 40 mg every other week or 20 mg weekly) the patient received, nor when the lymphoma was diagnosed relative to the start of treatment [19]. Westhovens et al. compared the effects of infliximab at different doses and found that a high dose of 10 mg/kg was associated with an increased risk of serious infections but not malignancy, including lymphoma [18]. Those significant limitations highlight the complexities in interpreting RCT-based data on the risk of lymphoma in anti-TNF-α therapy.

The observational studies demonstrated several strengths and limitations. A significant strength was the clear definition of the target populations. Two studies by Wolfe et al. [26,29] and Askling et al. [25,28] each involved large, diverse samples, which enhanced the external validity and generalizability of their results. The use of three comparator groups—general population, patients with RA on anti-TNF-α therapy, and patients with RA on methotrexate and/or placebo—provided a more significant assessment of the risk of lymphoma among recipients of anti-TNF-α. However, Geborek et al. reported a significantly higher rate of lymphoma in the anti-TNF-α group (RR = 11.5, 95% CI: 3.7–26.9); even so, the study’s small sample size and single-source population limited its internal validity [27]. The short follow-up period in the study may have impacted the results as well.

The studies generally varied in their methods of data collection, exposure times, and durations, all of which introduced variability. Wolfe et al. [26,29] provided detailed information on the incidence of lymphoma by type of anti-TNF-α therapy, which reduced the risk of misclassification bias. Conversely, other studies did not consistently report such information.

Large population samples, including those used by Askling et al. [25,28] and Wolfe et al. [26,29], pose challenges in collecting data about outcomes, which raises the risk of follow-up bias. Wolfe et al. relied on biannual questionnaires, which could introduce information, non-response, and reporting biases [26,29]. Conversely, Geborek et al. and Askling et al. utilized cancer registries, which might also introduce follow-up bias due to delays in registration [25,27,28].

Demographic variations, including age, sex, disease severity, and duration, additionally influence the incidence of lymphoma across intervention groups. For example, older female patients with longstanding disease showed a higher incidence of lymphoma [25,26], thereby highlighting the need for the careful consideration of those factors in interpreting the outcomes of research. Those demographic variations also emphasize the importance of tailoring study designs to account for such differences and thus ensuring more accurate and reliable results.

Overall, the event rate was low in all studies, which could increase the risk of statistical instability associated with having few events as well as limit the precision of pooled RRs. The issue of latency in diagnoses of lymphoma shortly after commencing anti-TNF-α treatment highlights the need for washout periods. For instance, Westhovens et al. [18] reported a diagnosis of lymphoma only 6 weeks after infliximab was begun, thereby suggesting a pre-existing subclinical condition. By contrast, Breedveld et al. [17] implemented a 4-week washout period before counting lymphoma outcomes, which reduced potential bias.

Furthermore, in Geborek et al.’s [27] study, accounting for latency periods (e.g., 6–12 months) significantly altered the estimated risks, thereby demonstrating the importance of considering latency in such analyses [33]. Those findings suggest that although anti-TNF-α therapies may not significantly increase the risk of lymphoma compared with conventional treatments, a careful consideration of study design and latency periods is crucial in interpreting the data. Those results underscore the need for ongoing vigilance and patient monitoring in clinical practice as well as the importance of robust study designs in future research.

### 4.2. Implications for Clinical Practice and Future Research

In clinical practice, it is essential for health care providers to engage in comprehensive patient counseling that emphasizes the overall safety profile of anti-TNF-α therapy. Standardized monitoring protocols should be implemented to ensure regular assessments in order to promptly detect any signs of lymphoma or other adverse effects. Future research should focus on prospective longitudinal studies with extended follow-up periods to elucidate the long-term safety profile of anti-TNF-α agents. Moreover, continued meta-analyses and comparative research on effectiveness are needed to refine estimated risks and compare outcomes with other treatment modalities. Exploring biological mechanisms and identifying biomarkers that predict the risk of lymphoma will further enhance current understandings of the lymphoma risk in RA patients treated with anti-TNF-α agents and inform personalized treatment approaches for patients with RA.

## 5. Conclusions

This systematic review and meta-analysis provides important insights into the risk of lymphoma associated with anti-TNF-α therapy in patients with RA. The analysis of the included studies indicated no statistically significant difference in the risk between patients treated with anti-TNF-α agents and those receiving conventional therapies. However, findings from observational studies exhibit considerable heterogeneity that suggests potential variability in the risk of lymphoma across different patient populations and treatment settings. Although some studies have hinted at a possible elevated risk of lymphoma in specific subgroups, the overall evidence does not support a definitive conclusion regarding any such risk with anti-TNF-α therapy. Future research should prioritize longitudinal studies with extended follow-up in order to elucidate the long-term safety profile of anti-TNF-α therapy. Continued investigation into potential risk factors and refined comparative analyses of effectiveness are also crucial for informing clinical decision-making and optimizing treatment strategies in the management of RA. 

## Figures and Tables

**Figure 1 medicina-60-01156-f001:**
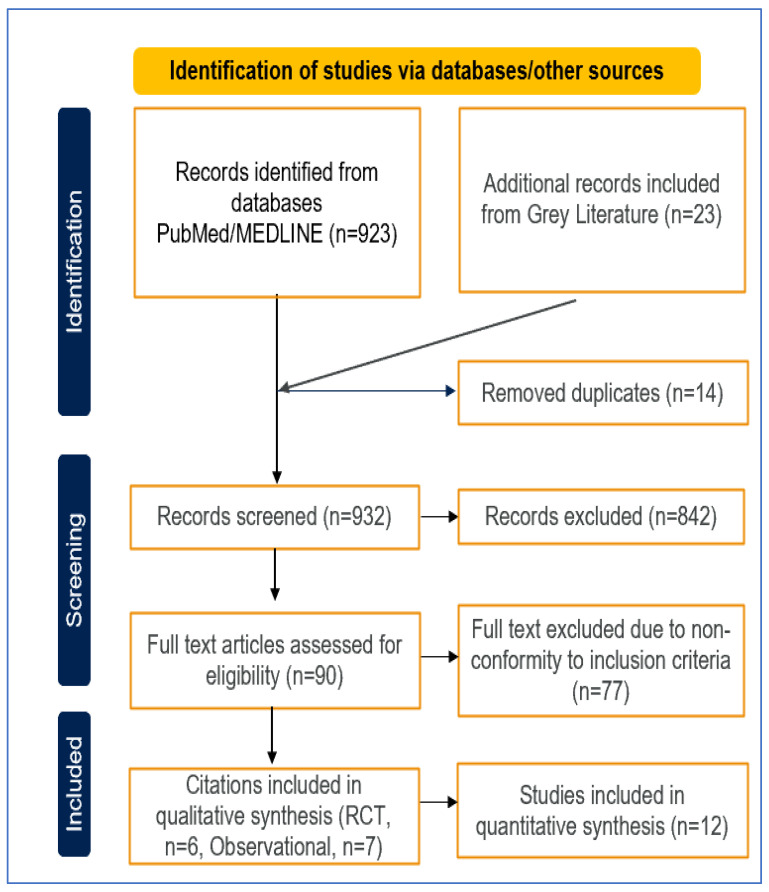
PRISMA checklist for identified studies. Figure 1 presents the Preferred Reporting Items for Systematic Reviews and Meta-Analyses (PRISMA) checklist for the identification and selection of the literature included in the systematic review and meta-analysis. The checklist outlines the key stages of the review, including the identification, screening, eligibility, and inclusion of the literature based on predetermined criteria.

**Figure 2 medicina-60-01156-f002:**
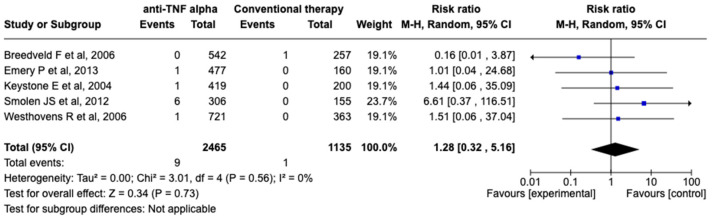
Risk ratios (RRs) and forest plot for RCTs [17,18,19,21,22]. Figure 2 presents the forest plot depicting RRs derived from RCTs included in the meta-analysis. Each study’s estimated RR is represented by a square that is proportional to the study’s weight in the meta-analysis, with horizontal lines indicating the 95% CI. The diamond at the bottom represents the overall pooled RR, which provides insight into the cumulative effect of anti-TNF-α therapy on the risk of lymphoma in patients with RA based on data from RCTs. The plot offers a visual summary of the estimated RRs and their variability across RCTs, which can guide the assessment of treatment efficacy and safety.

**Figure 3 medicina-60-01156-f003:**
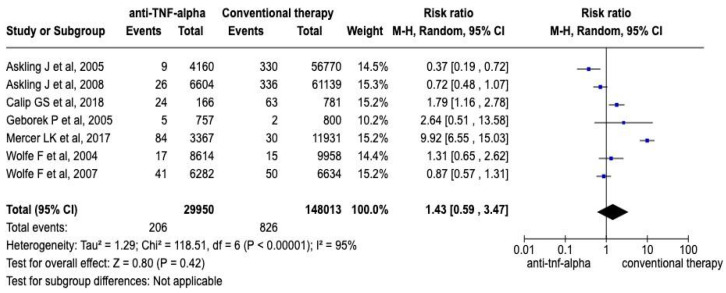
RRs and forest plot for observational studies [23,24,25,26,27,28,29]. Figure 3 displays a forest plot illustrating RRs derived from observational studies included in the meta-analysis. Each study’s estimated RR is represented by a square, sized to reflect the study’s weight in the meta-analysis. Horizontal lines indicate the 95% CI for each study’s estimated RR. The diamond at the bottom of the plot represents the overall pooled RR, with a width indicating the corresponding 95% CI. The plot visually represents the variability in estimated RRs across studies and the overall effect size of anti-TNF-α therapy on the risk of lymphoma in patients with RA based on data from observational studies.

**Figure 4 medicina-60-01156-f004:**
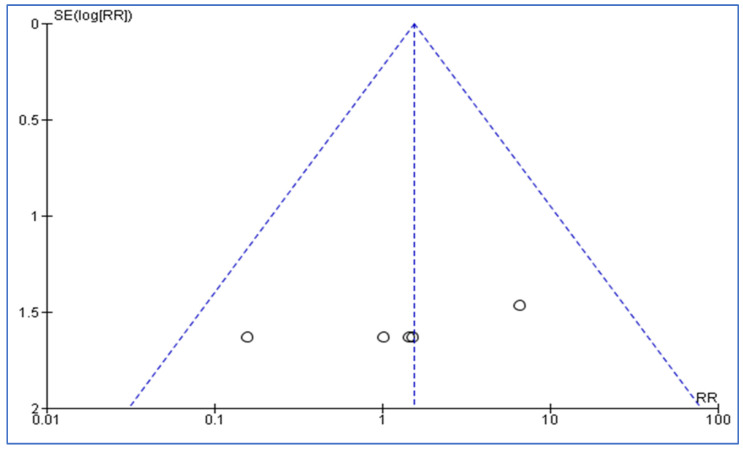
Funnel plot for RCTs reviewed. Figure 4 illustrates a funnel plot analysis for RCTs reviewed in the meta-analysis. The plot is symmetrical, with results widely distributed within the funnel, which indicates a lack of publication bias among the RCTs reviewed.

**Figure 5 medicina-60-01156-f005:**
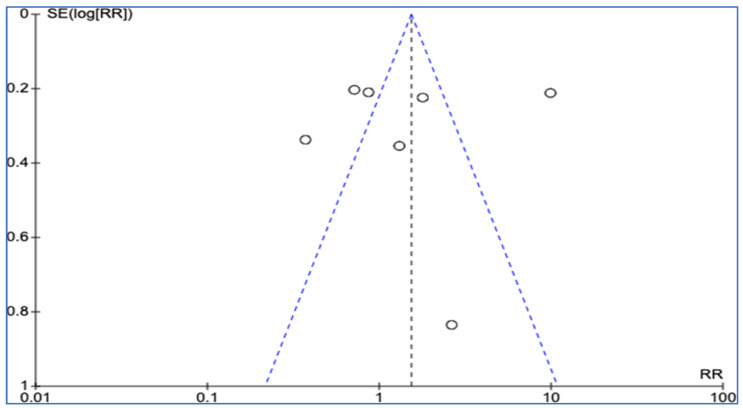
Funnel plot for observational studies. Figure 5 presents a funnel plot analysis for the observational studies reviewed in the meta-analysis. The plot is asymmetrical, with four results lying outside the plot, which suggests potential publication bias among the observational studies reviewed.

**Figure 6 medicina-60-01156-f006:**
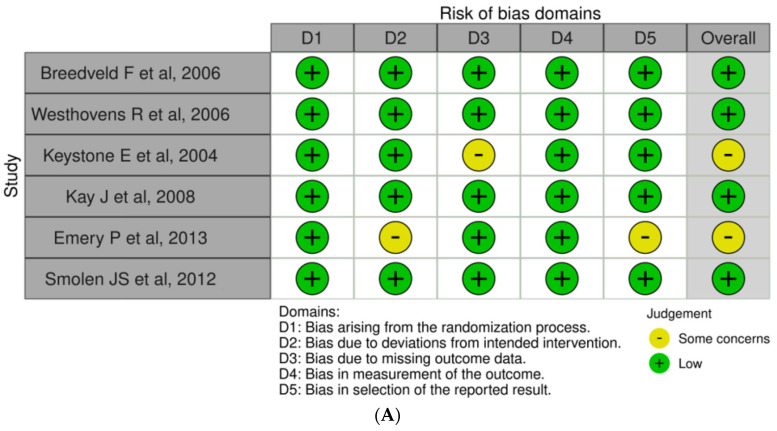
(**A**) Traffic light plots of the domain-level judgments for the results of each study [17,18,19,20,21,22]. (**A**) displays traffic light plots depicting domain-level judgments for the results of each study. The assessments highlight variations in risk-of-bias across different domains. (**B**) Weighted bar plots of the distribution of risk-of-bias judgments within each bias domain. (**B**) presents weighted bar plots illustrating the distribution of risk-of-bias judgments within each bias domain. Studies are categorized based on their risk-of-bias assessments across multiple domains.

**Table 1 medicina-60-01156-t001:** Characteristics of randomized controlled trials (RCTs) reviewed.

Study Number	1	2	3	4	5	6
Authors	Breedveld et al. (2006) [17]	Westhovens et al. (2006) [18]	Keystone et al. (2004) [19]	Kay et al. (2008) [20]	Emery et al. (2013) [21]	Smolen et al. (2012) [22]
Location	Australia, Europe, and North America	US	US	US	UK	US
Total RA population	799	1084	619	172	637	461
Anti-TNF population	542	721	419	137	477	206
Conventional treatment population	257	363	200	35	160	155
Type of anti-TNF-α agent	Adalimumab	Infliximab + MTX 3 mg/kg; Infliximab + MTX 10 mg/kg	Adalimumab	Golimumab	Golimumab 100 mg + placebo Golimumab 50 mg + MTX Golimumab 100 mg + MTX	Golimumab 100 mg Golimumab 50 mg
Anti-TNF-α treatment duration (weeks)	96	46	52	48	104	160
Type of conventional treatment	Methotrexate + placebo	Methotrexate + placebo	Methotrexate + placebo	Methotrexate + placebo	Methotrexate + placebo	Placebo
Incidence of lymphoma in anti-TNF group	0	1	1	0	1	6
Incidence of lymphoma in conventional group	1	0	0	0	0	0
Efficacy and safety summary points	Combination therapy with adalimumab plus MTX was significantly superior to MTX or adalimumab alone in early, aggressive RA, improved signs and symptoms, and inhibited radiographic progression. Adverse event profiles were comparable.	Induction and maintenance regimen of 3 mg/kg infliximab plus MTX was tolerated well. The 10 mg/kg infliximab induction regimen with MTX had an increased risk of serious infection and a risk similar to MTX alone.	Adalimumab significantly inhibited structural joint damage, improved symptoms and function in patients with RA with inadequate MTX response and was tolerated well with fewer discontinuations than the placebo.	Golimumab plus MTX achieved ACR20 response significantly more than placebo plus MTX. Clinical responses, observed as early as Week 2, persisted through Week 52.	Golimumab plus MTX yielded significant ACR20 and ACR50 responses, HAQ DI improvements, and less radiographic progression than MTX alone or golimumab monotherapy up to 2 years.	Golimumab 50 mg and 100 mg injections every 4 weeks sustained improvements in RA symptoms and function in most patients and had safety consistent with other TNF inhibitors.

Abbreviations: MTX: methotrexate, RA: rheumatoid arthritis, ACR20: American College of Rheumatology 20% improvement criteria, ACR50: American College of Rheumatology 50% improvement criteria, HAQ DI: Health Assessment Questionnaire Disability Index, TNF: tumor necrosis factor, US: United States, UK: United Kingdom.

**Table 2 medicina-60-01156-t002:** Characteristics of observational studies reviewed.

Study Number	1	2	3	4	5	6	7
Authors	Calip et al. (2018) [23]	Mercer et al. (2017) [24]	Askling et al. (2008) [25]	Wolfe et al. (2007) [26]	Geborek et al. (2005) [27]	Askling et al. (2005) [28]	Wolfe et al. (2004) [29]
Country	US	UK	Sweden	US	Sweden	Sweden	US
Total RA population	947	15,298	67,743	12,916	1557	60,930	18,572
Anti-TNF population	166	3367	6604	6282	757	4160	8614
Conventional treatment population	781	11,931	61,139	6634	800	56,770	9958
Type of anti-TNF-α agent	Infliximab, adalimumab, etanercept, golimumab, and certolizumab pegol	Infliximab, adalimumab, and etanercept	Infliximab, adalimumab, and etanercept	Infliximab, etanercept, and adalimumab	Etanercept or infliximab	Infliximab, adalimumab, and etanercept	Infliximab and etanercept
Anti-TNF-α treatment duration (Years)	<1.5 to >2.5	NA	NA	3	1.7	NA	NA
Type of conventional treatment	TNFI non-users	csDMARD	Anti-TNF-naïve patients with RA	Placebo group	Conventional antirheumatic treatment	RA patients enrolled in inpatients and early arthritis registries	MTX without biologic No MTX or biologic
Incidence of lymphoma in anti-TNF group	24	84	26	41	5	9	17
Incidence of lymphoma in conventional group	63	30	336	50	2	330	15
Follow-up	84	90.6	96	48	19	36	16
Efficacy and safety summary points	Cases of NHL had greater TNFI use than controls (33% vs. 20%). TNFI use was linked to a nearly twofold increased risk of NHL (OR = 1.93; 95% CI: 1.16–3.20). TNF fusion protein (i.e., etanercept) was linked to an increased risk of NHL (OR = 2.73; 95% CI: 1.40–5.33) but not anti-TNF monoclonal antibodies (OR = 1.77; 95% CI: 0.87–3.58).	When 11,931 TNFI-treated patients were compared with 3367 biologic-naïve patients; the risk of lymphoma between TNFI and biologic-naïve groups after adjustment did not differ (HR = 1.00; 95% CI: 0.56–1.80). No differences in risk were observed for individual TNFI.	Across 26,981 person-years, 6604 anti-TNF-treated patients with RA had 26 lymphomas (RR = 1.35; 95% CI: 0.82–2.11) versus anti-TNF-naive patients with RA (RR = 2.72; 95% CI: 1.82–4.08) versus the general population. Early starters (i.e., 1998–2001) had an elevated risk of lymphoma. No significant variation according to treatment duration or type emerged.	The risk of non-melanotic skin cancer (OR = 1.5; 95% CI: 1.2–1.8) and melanoma (OR = 2.3; 95% CI: 0.9–5.4) rose with biologics. No other malignancy was linked to biologic use. Overall, the risk of cancer’s OR was 1.0 (95% CI: 0.8–1.2).	The anti-TNF group had 16 tumors (i.e., 5 lymphomas) across 1603 person-years, while the comparison group had 69 tumors (i.e., 2 lymphomas) in 3948 person-years. Lymphoma’s RR was 11.5 (95% CI: 3.7–26.9) for anti-TNF and 1.3 (95% CI: 0.2–4.5) for the comparison group.	The prevalence and incidence of patients with RA were associated with increased risks of lymphoma (SIR = 1.9 and 2.0, respectively) and leukemia (SIR = 2.1 and 2.2, respectively). TNF antagonist-treated patients had triple the risk of lymphoma (SIR = 2.9) versus the general population, but not a higher risk versus other cohorts with RA.	Overall, the SIR for lymphoma was 1.9 (95% CI: 1.3–2.7), for biologic use was 2.9 (95% CI: 1.7–4.9), for infliximab (i.e., with or without etanercept) was 2.6 (95% CI: 1.4–4.5), for etanercept (i.e., with or without infliximab) was 3.8 (95% CI: 1.9–7.5), and for MTX was 1.7 (95% CI: 0.9–3.2).

Abbreviations: TNFI: tumor necrosis factor inhibitor, NHL: non-Hodgkin lymphoma, OR: odds ratio, CI: confidence interval, NA: not applicable, csDMARD: conventional synthetic disease-modifying anti-rheumatic drug, HR: hazard ratio, RR: relative risk, SIR: standardized incidence ratio, MTX: methotrexate, RA: rheumatoid arthritis.

## Data Availability

The raw data supporting the conclusions of this article will be made available by the author on request.

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
