# Peer review of "Anti-TNF Alpha and Risk of Lymphoma in Rheumatoid Arthritis: A Systematic Review and Meta-Analysis"

_medicina, 2024, doi:10.3390/medicina60071156_

Round 1

Reviewer 1 Report

Comments and Suggestions for Authors

The paper approaches a debated topic, still controversal, depending on the reported prevalence. The resources are valuable and extensively presented.The review pointed out that concerns regarding the risk of lymphoma should not raise difficulties in choosing a therapeutic management when treating RA patients.

Moderate english editing is required, as well as a more clear presentation of the valuable data.

Comments on the Quality of English Language

Some phrases are difficult to understand and it might interfere with information processing. It should be revised by an experienced english editor in order to be presented in a proper manner, enabling a valuable research for future readers.

Reviewer 2 Report

Comments and Suggestions for Authors

The manuscript discusses the association of anti-TNF treatment with lymphoma. It is an interesting study, however there are minor issues that can be improved:

  1. Authors should add the information on TNF biology in the Introduction section and the information on lymphoma. There are different types of lymphoma and it is not clear which lymphoma authors refer to. 
  2. Some sentences lack references. Authors should proof-read and add references throughout the manuscript 
  3. If some figures are not original, authors should add the references and some explanation under all figures. 

Reviewer 3 Report

Comments and Suggestions for Authors

Font of Tables is unreadable and the organization of tables have to be re-distrubuted

Figure 2 font is cropped and strentched with awful resolution

All figures have to be summplemented with legends not only figure titles

The manuscript includes several typo and grammatical mistakes that have to be revised carefully 

The conclusion and future recommendations have to be re-written with a more visionary prespective

Comments on the Quality of English Language

The manuscript includes several typo and grammatical mistakes that have to be revised carefully 

Round 2

Reviewer 3 Report

Comments and Suggestions for Authors

ALL COMMMENTS ARE TACKLED 

Comments on the Quality of English Language

Accepted